# Global DNA Methylation in Poorly Controlled Type 2 Diabetes Mellitus: Association with Redox and Inflammatory Biomarkers

**DOI:** 10.3390/ijms26146716

**Published:** 2025-07-13

**Authors:** Sanja Vujcic, Jelena Kotur-Stevuljevic, Zoran Vujcic, Sanja Stojanovic, Teodora Beljic Zivkovic, Miljanka Vuksanovic, Milica Marjanovic Petkovic, Iva Perovic Blagojevic, Branka Koprivica-Uzelac, Sanja Ilic-Mijailovic, Manfredi Rizzo, Aleksandra Zeljkovic, Tatjana Stefanovic, Srecko Bosic, Jelena Vekic

**Affiliations:** 1Department of Medical Biochemistry, University of Belgrade-Faculty of Pharmacy, 11000 Belgrade, Serbia; sanja.vujcic@icloud.com (S.V.); jelena.kotur@pharmacy.bg.ac.rs (J.K.-S.); aleksandra.zeljkovic@pharmacy.bg.ac.rs (A.Z.); 2Department of Biochemistry, University of Belgrade-Faculty of Chemistry, 11000 Belgrade, Serbia; zvujcic@chem.bg.ac.rs; 3Department of Chemistry, University of Belgrade-Institute of Chemistry, Technology and Metallurgy, 11000 Belgrade, Serbia; sanja.stojanovic@ihtm.bg.ac.rs; 4Department of Internal Medicine, University of Belgrade-Faculty of Medicine, 11000 Belgrade, Serbia; beljicdora@gmail.com (T.B.Z.); miljankavuk@gmail.com (M.V.); drmilicamarjanovic@gmail.com (M.M.P.); 5Division of Endocrinology, Diabetes and Metabolic Disorders, Department of Internal Medicine, Zvezdara University Medical Center, 11000 Belgrade, Serbia; 6Clinical Hospital Center “Dr. Dragisa Misovic-Dedinje”, 11000 Belgrade, Serbia; ivapb17@gmail.com (I.P.B.); bkoprivica@beotel.net (B.K.-U.); ilicsanja1969@yahoo.com (S.I.-M.); 7Department of Health Promotion, Mother and Child Care, Internal Medicine and Medical Specialties, University of Palermo, 90100 Palermo, Italy; 8General Hospital Pozarevac, 12000 Pozarevac, Serbia; tatjanastefanovicpo@gmail.com (T.S.); delboske2@gmail.com (S.B.)

**Keywords:** global DNA methylation, biomarkers, diabetes mellitus, oxidative stress, inflammation

## Abstract

Although emerging evidence suggests that epigenetic mechanisms contribute to the pathogenesis and progression of type 2 diabetes mellitus (T2DM), data remain limited for patients with suboptimal metabolic control. The aim of this study was to assess global DNA methylation in patients with poorly controlled T2DM and to identify diabetes-related factors associated with DNA methylation levels. The study included 107 patients and 50 healthy controls. Global DNA methylation (5mC) was measured by UHPLC-DAD method. Pro-oxidant and antioxidant biomarkers, advanced glycation end-products, high-sensitivity C-reactive protein (hsCRP) and complete blood count were determined and leukocyte indices calculated. Patients had a significantly lower 5mC than controls (3.56 ± 0.31% vs. 4.00 ± 0.68%; *p* < 0.001), with further reductions observed in those with longer disease duration and diabetic foot ulcers. Oxidative stress and inflammatory biomarkers were higher in the patient group. DNA hypomethylation was associated with a higher monocyte-to-lymphocyte ratio and hsCRP, pro-oxidant–antioxidant balance, ischemia-modified albumin, and advanced oxidation protein products levels. Conversely, 5mC levels showed positive correlations with total antioxidant status and total sulfhydryl groups. Principal component analysis identified five key factors: proinflammatory, pro-oxidant, aging, hyperglycemic, and antioxidant. The pro-oxidant factor emerged as the sole independent predictor of global DNA hypomethylation in T2DM (OR = 2.294; *p* = 0.027). Our results indicate that global DNA hypomethylation could be a biomarker of T2DM progression, reflecting the complex interactions between oxidative stress, inflammation, and epigenetic modifications in T2DM.

## 1. Introduction

Type 2 diabetes mellitus (T2DM) and its related complications continue to pose a major public health challenge [1]. Recent studies emphasize the role of epigenetic dysregulation in T2DM, particularly in the development of microvascular and macrovascular complications [2]. One of the most studied epigenetic modifications is DNA methylation, in which a methyl group is added to the 5-carbon of cytosine in CpG dinucleotides. This process is catalyzed by DNA methyltransferases (DNMTs) [3]. DNA methylation is considered a crucial epigenetic mechanism due to its role in regulating gene expression, including genes involved in glucose [4] and lipid metabolism [5].

Oxidative stress and inflammation are frequently present in poorly controlled T2DM [6]. Novel findings suggest that these factors may cause changes in DNA methylation [7]. Chronic hyperglycemia through several metabolic pathways contributes to the overproduction of reactive oxygen species (ROS), which can overcome the body’s own antioxidant defense, leading to oxidative stress [8]. Additionally, persistent hyperglycemia leads to the formation of advanced glycation end-products (AGEs), which may further increase ROS production through their interaction with receptors for AGEs (RAGE) [9]. Glycoxidative modifications of proteins such as IgG have been implicated in the immunopathology of T2DM, supporting the pathological relevance of AGEs in the disease process [10]. It has been established that ROS may affect DNA methylation through direct DNA damage or interaction with DNMTs [11]. Although a direct link between AGEs and DNA methylation has not yet been fully established, it may involve common pathways. Both oxidative stress and AGEs promote the production of proinflammatory cytokines and activate the nuclear factor kappa-B (NF-κB) pathway, which can alter the activity of DNMTs, leading to changes in DNA methylation [12]. These epigenetic alterations can further affect genes involved in inflammation and the oxidative stress response, creating a feedback loop that accelerates the progression of T2DM.

Laboratory assessment of global DNA methylation, by estimating the 5-methylcytosine (5mC) percentage in total cytosine within DNA, could be a potentially useful epigenetic biomarker for screening DNA methylation status. While it is known that hypermethylation of gene promoters leads to their silencing, the effects of alterations in global DNA methylation on the pathophysiology of complex diseases, particularly T2DM, are not yet fully understood [13]. Previous studies have reported conflicting results, finding both global DNA hypomethylation [14] and hypermethylation [15] in T2DM. However, data are limited, especially regarding DNA methylation in patients with poorly controlled diabetes. This study focuses on its interplay with oxidative stress and inflammation, aiming to address the still-existing gap in the current understanding of the role of DNA methylation in the pathophysiology of T2DM.

The aim of this study was to examine DNA methylation patterns in patients with poorly controlled T2DM. Specifically, we aimed to determine whether significant differences in DNA methylation levels exist between patients with and without diabetes-related complications and to identify the diabetes-related factors that influence global DNA methylation levels.

## 2. Results

The demographic and laboratory data of the T2DM patients and the control group (CG) are summarized in Table 1. The patients were older and had a higher proportion of males. In addition, T2DM patients had higher systolic blood pressure and prevalence of hypertension, as well as a higher body mass index (BMI) and prevalence of obesity, as compared to the CG. The proportion of smokers was also higher in the T2DM group. As expected, the patients had significantly higher levels of glucose and glycated hemoglobin (HbA_1c_). The analysis of lipid status showed that the patients had higher triglycerides (TG), but lower levels of total cholesterol (TC), high-density lipoprotein cholesterol (HDL-C), and low-density lipoprotein cholesterol (LDL-C) than the controls.

In our T2DM group, the median duration of diabetes was 10 years (interquartile range: 3–15 years), with approximately one-third (35.5%) of patients having had T2DM for more than 10 years. Regarding complications of diabetes, 48 patients (44.9%) had cardiovascular disease (CVD), and 66 patients (58.9%) had microvascular complications. Neuropathy was present in 51 (47.7%), retinopathy in 8 (7.5%), and nephropathy in 17 (15.9%) patients, while 25 patients (23.4%) had diabetic foot ulcers. Almost a third of our patients (39.9%) experienced both microvascular and macrovascular complications.

The levels of global DNA methylation in T2DM patients and CG are shown in Figure 1. T2DM patients had significantly lower 5mC levels (3.56 ± 0.31%) than controls (4.00 ± 0.68%; *p* < 0.001). The observed difference persisted even after adjustment for age and gender (adjusted means for 5mC: 3.57 ± 0.45% in T2DM vs. 3.98 ± 0.55% in CG; *p* < 0.001).

A correlation analysis showed that the global DNA methylation level was inversely associated with age (r = −0.213; *p* = 0.009) and serum glucose concentration (r = −0.388; *p* < 0.001). On the other hand, a positive association was found between global DNA methylation and HDL-C level (r = 0.228; *p* = 0.008).

We further analyzed whether the level of global DNA methylation differs depending on the clinical characteristics of the T2DM patients studied (Table 2). This analysis showed that patients with a T2DM duration of ≥10 years had significantly lower global DNA methylation levels than those who had diabetes for less than 10 years. Reduced global DNA methylation was also observed in T2DM patients who smoke, although the difference did not reach statistical significance. Regarding T2DM complications, there was no difference in 5mC levels between patients with and without CVD. However, a trend towards reduced global DNA methylation was observed in patients with microvascular complications. Patients with diabetic foot ulcers exhibited significantly lower global DNA methylation levels than patients without this complication.

In order to elucidate factors affecting global DNA methylation, we first analyzed the levels of F-AGEs in both groups. The results showed that F-AGEs levels in T2DM patients (median: 5.44 U/mL, Q1–Q3: 4.91–7.71 U/mL) were comparable to those of CG (median: 5.82 U/mL, Q1–Q3: 4.57–6.71 U/mL; *p* = 0.139). There was no significant correlation between global DNA methylation and F-AGEs levels (r = 0.060; *p* = 0.478).

Next, we analyzed inflammatory and redox biomarkers in T2DM patients and controls (Table 3). Inflammatory markers such as high-sensitivity C-reactive protein (hsCRP) and blood-count derived biomarkers (neutrophil-to-lymphocyte ratio (NLR), monocyte-to-lymphocyte ratio (MLR), monocyte-to-white blood cell ratio (MWR)) were significantly elevated in T2DM patients. Pro-oxidant biomarkers (total oxidant status (TOS) and pro-oxidant-antioxidant balance (PAB)) and biomarkers of oxidative damage (ischemia-modified albumin (IMA) and advanced oxidation protein products (AOPP)) were significantly increased in T2DM patients (*p* < 0.001), while antioxidant defense biomarkers (total antioxidant status (TAS), total sulfhydryl (SH) groups) were significantly reduced (*p* < 0.001).

Further analysis was performed to investigate the correlations of 5mC levels with inflammatory and redox biomarkers (Table 4). It was found that the 5mC levels were significantly negatively correlated with hsCRP and MLR. Regarding redox biomarkers, 5mC levels exhibited significant negative correlations with PAB, IMA, and AOPP. Conversely, 5mC levels showed positive correlations with the levels of antioxidative defense biomarkers TAS and SH-groups.

Multiple linear regression analysis with stepwise selection was performed to identify potential independent associations between global DNA methylation and variables that showed significant correlations with 5mC levels in univariate analysis. In particular, the following variables were included in the model: age, gender, glucose, hsCRP, MLR, PAB, IMA, AOPP, TAS, and SH-groups. In the multivariate analysis, global DNA methylation level remained significantly inversely associated with age (β = −0.254; *p* = 0.004), while showing positive associations with the antioxidant biomarkers TAS (β = 0.342; *p* < 0.001) and SH-groups (β = 0.204; *p* = 0.023).

To test which of the investigated variables are possible independent predictors of DNA hypomethylation in T2DM, we first reduced the number of parameters by principal component analysis (PCA). The factors extracted by PCA analysis were composed of parameters with a comparable level of variability. The Kaiser–Meyer–Olkin index, as a measure of sampling adequacy, was 0.562 (fulfilling the condition of KMO >0.500), and the Bartlett’s test of sphericity, which yielded *p* = 0.001 (the condition for the Bartlett’s test *p* is below 0.05), established the analysis’s significance. The extracted factors with loadings of the included variables are presented in Table 5.

The factors extracted in the PCA analysis explained 58% of the variance of the included parameters. The first factor was characterized by positive loadings for hsCRP, NLR, PAB and IMA, and a negative loading for superoxide dismutase (SOD) activity. It was interpreted as the “proinflammatory factor” and explained 18% of the total variance. The second factor explained 12.5% of the total variance and was denoted as the “pro-oxidant factor” due to positive loadings for TOS, AOPP levels and BMI. The third factor captured 10.5% of the total variance and was named the “ageing factor” due to positive loadings for patients’ age and diabetes duration. The fourth factor was termed the “hyperglycemic factor,” and was characterized by positive loadings for F-AGEs and gender and negative loading for glucose concentration. This factor explained 9% of the total variance. The fifth factor included SH-groups with a positive loading, and was therefore labelled the “antioxidant factor”, which explained 8% of the total variance.

Binary logistic regression analysis with the PCA-derived factor scores as independent variables was used to assess their potential independent association with DNA hypomethylation (5mC < 3.40%). As shown in Table 6, among the extracted factors, the sole significant predictor of DNA hypomethylation was the “pro-oxidant factor”. Specifically, an increased “pro-oxidant factor” significantly increased the probability of low 5mC levels (OR= 2.294; *p* = 0.027).

## 3. Discussion

In the presented study, we have demonstrated that poorly controlled T2DM patients have decreased global DNA methylation. In addition, our data suggest a further decrease in global DNA methylation levels with increasing age and duration of diabetes, as well as in the presence of microvascular complications, particularly diabetic foot ulcers. Furthermore, significant associations were observed between global DNA methylation and biomarkers of inflammation and redox status, while the “pro-oxidant factor” emerged as the only significant predictor of DNA hypomethylation in poorly controlled T2DM.

DNA methylation is a fundamental epigenetic modification in which a methyl group is added to DNA, typically to cytosine in CpG dinucleotides, leading to changes in gene expression without altering the DNA sequence [3]. In T2DM, aberrant DNA methylation patterns have been observed in genes involved in glucose metabolism and insulin signaling pathways [16,17]. These epigenetic changes can be further influenced by oxidative stress and chronic inflammation, which typically occur in poorly controlled T2DM. DNA methylation, in addition to serving as a link between environmental factors and genetic predisposition to T2DM, could also be a valuable biomarker for monitoring disease progression. However, while global DNA methylation has been extensively studied in cancer, studies in T2DM have been very limited, and the reported results are conflicting. In the current study, T2DM patients had significantly lower 5mC levels compared to control subjects (Figure 1). Consistent with our results, Luttmer et al. [14] found DNA hypomethylation in individuals with high fasting plasma glucose levels. In addition, Thongsroy et al. [18] found a decrease in DNA methylation with increased HbA_1c_ levels. In contrast, Pinzón-Cortés et al. [15] reported no difference in 5mC levels in T2DM patients compared to healthy controls, but found hypermethylation in poorly regulated T2DM. Recently, Muka et al. [19] conducted a systematic review, and established an inconsistent association between global DNA methylation and T2DM, highlighting the current lack of studies in this area, and the need for further research.

One of the possible reasons for such inconsistencies could be the use of different methods for assessing global DNA methylation across studies. While the HPLC method is considered the gold standard, other techniques such as ELISA, Alu, and LUMA are commonly used due to their simplicity and cost-effectiveness [20]. However, the use of different methodological approaches complicates result interpretation and limits the comparability of findings across studies, indicating the need for harmonization. Furthermore, discrepancies in the results may also be influenced by the characteristics of the patients included in the studies. For instance, our patients were older and had higher HbA_1c_ levels than the patients included in the study by Pinzón-Cortés et al. [15]. In addition, our study had a higher proportion of male patients and a greater obesity rate in the T2DM group. Of note, although our patients were older and included a higher proportion of males compared to the CG, the observed difference in global DNA methylation levels remained significant even after adjustment for age and gender. In the current study, we did not perform a gender-specific analysis, despite a significant difference in the gender distribution among participants. Future studies with larger and more balanced cohorts are needed to explore potential sex-specific effects on global DNA methylation and its association with oxidative stress and inflammation.

Analysis of global DNA methylation in relation to the clinical characteristics of our patients revealed that patients with a longer duration of diabetes and those with diabetic foot ulcers had lower global DNA methylation levels (Table 2), suggesting that chronic hyperglycemia and severe complications might exacerbate epigenetic changes, and vice versa. Current data on patients with poorly controlled T2DM and developed complications are very limited. Maghbooli et al. [21] found that global DNA methylation is a potential biomarker of diabetic retinopathy, by comparing global DNA methylation between T2DM patients with and without retinopathy. In this study, however, no difference in 5mC levels was found between patients with and without retinopathy, likely due to the small number of patients with this microvascular complication. Zhang et al. [22] found a decrease in DNA methylation in diabetic neuropathy. Consistent with this finding, a trend towards decreased 5mC levels was observed in our patients with neuropathy, while it was significantly decreased in patients with diabetic foot ulcers. Additionally, we detected a trend towards global DNA hypomethylation in patients with nephropathy, although the difference did not reach statistical significance. Although meaningful trends were observed, the relatively small number of participants in certain subgroups, particularly those with diabetic retinopathy, may have limited the ability to detect statistically significant associations. As such, some findings should be interpreted with caution, bearing in mind the limited statistical power rather than the lack of a true association. Future studies with larger and more evenly distributed patients are needed to confirm and expand upon these observations.

Studies investigating the link between global DNA methylation and CVD risk have reported significant associations with both increased [23] and decreased [24] global DNA methylation levels. It is important to note that global hypomethylation has been specifically associated with atherosclerotic plaques [25]. Wang et al. [26] found global hypomethylation and the subsequent upregulation of genes implicated in cytokine–cytokine receptor interactions and the MAPK signaling pathway. These molecular changes promote inflammatory responses by recruiting immune cells, thereby contributing to both plaque formation and instability. Additionally, activation of the MAPK signaling pathway by cytokines and stress signals facilitates smooth muscle cell proliferation, macrophage foam cell formation, and apoptosis, processes that drive plaque progression and increase the risk of rupture. Similarly, Zuo et al. [27] demonstrated a correlation between hypomethylation of the interleukin-6 promoter and the development of coronary artery disease. In the current study no significant differences in 5mC levels were observed in relation to the presence of CVD. A possible explanation could be that DNA methylation is affected by both genetic and environmental factors, which could contribute to the inconsistency in the association with CVD observed in the studies. Patient-related factors, such as lifestyle habits, medication use and other comorbidities might also contribute to inconsistent results among studies. Furthermore, since CVD encompasses a broad spectrum of conditions, changes in DNA methylation could occur at earlier stages of disease development, making it difficult to detect differences at a single time point. Finally, global DNA methylation may vary between tissues. In the current study, peripheral blood mononuclear cells (PBMCs) were used, as liquid-biopsy is considered the most convenient minimally invasive procedure for obtaining biological samples for DNA methylation analysis [28].

One of the known consequences of chronic hyperglycemia is the accumulation of AGEs and increased expression of RAGE, whose interaction activates the NF-κB signaling pathway [9]. This leads to the production of proinflammatory and prothrombotic factors and further ROS generation, which promotes angiogenesis, oxidative stress, and cell proliferation, thereby contributing to the development of microvascular and macrovascular complications [29]. In the current study, no significant difference in F-AGEs levels was observed between T2DM patients and controls, and there was no significant correlation between global DNA methylation and AGEs levels. Previous studies have emphasized the role of AGEs in the pathogenesis of diabetic complications [30,31], but their direct impact on DNA methylation remains unclear. At this point, it should be mentioned that the method used for the assessment of AGEs has certain limitations, as it only detects fluorescent AGEs, which represent a smaller fraction of the total AGEs in serum. Furthermore, AGEs in serum represent only a part of the body’s AGE pool, which includes both circulating AGEs and AGEs accumulated in tissues, whose contribution remains unknown [31]. Our finding of an inverse correlation between 5mC levels and both age and serum glucose concentration (Table 3) suggests that advanced age and poor glycemic control may contribute to altered DNA methylation patterns. However, although PCA analysis (Table 5) extracted “ageing” and “hyperglycemic” factors, including glucose and F-AGEs levels, these factors were not shown to be independent predictors of global DNA hypomethylation in this patient group (Table 6). This brings to attention the need to further investigate the role of glucose itself and F-AGEs on DNA methylation in patients with poorly controlled diabetes.

Another feature of poorly regulated T2DM is the presence of oxidative stress and inflammation (Table 2), both of which can promote DNA hypomethylation by affecting key enzymes involved in epigenetic regulation, namely DNMTs and ten-eleven translocation (TET) enzymes [32]. DNMTs are responsible for maintaining established DNA methylation patterns and adding new methyl groups, a process known as de novo methylation. In contrast, TET enzymes catalyze the reverse process, by initiating DNA demethylation through the oxidation of 5mC to 5-hydroxymethylcytosine (5hmC), ultimately resulting in the demethylation of cytosine. Since molecular oxygen is required for TET activity, oxidative stress could increase TET activity due to the increased bioavailability of oxygen, leading to the production of 5hmC and subsequent DNA demethylation [33]. In addition, ROS inhibit DNMT activity by directly oxidizing the enzyme, altering its structure and function, and damaging DNA, which prevents proper DNMT binding [34]. Besides, ROS can interfere with the availability of the essential cofactor S-adenosylmethionine (SAM), which serves as a methyl group donor, further reducing DNA methylation. Indeed, multivariate regression analysis identified older age and exhausted antioxidative defense mechanisms as significant independent predictors of reduced global DNA methylation levels in our T2DM patients. More importantly, PCA analysis has identified the “pro-oxidant factor” as the only independent predictor of DNA hypomethylation (Table 6). These associations suggest that exacerbated OS is a major driver of aberrant DNA methylation in poorly controlled T2DM.

Our findings contribute to the growing body of evidence linking oxidative stress and inflammation to global DNA methylation alterations in T2DM. In our cohort, patients with poorly controlled T2DM exhibited significantly disturbed redox balance, as reflected by increased TOS, PAB, IMA, and AOPP levels, alongside decreased TAS and SH-groups (Table 3). As previously described, oxidative stress can affect global DNA methylation through both direct DNA damage and modulation of DNMT and TET enzymes [32,33]. In support of this, we observed negative correlations between 5mC levels and pro-oxidant biomarkers, and positive correlations with antioxidative defense biomarkers (Table 4), suggesting redox-influenced regulation of global DNA methylation. Inflammation also appears to influence the observed epigenetic patterns. Previous studies have shown that inflammatory cytokines such as TNF-α and IL-6 can modulate DNMT expression via NF-κB signaling, ultimately leading to global DNA hypomethylation [35]. Moreover, chronic inflammation, particularly TNF-α, contributes to ROS production, thereby indirectly promoting DNA hypomethylation [36]. In our study, hsCRP and MLR were significantly elevated in the T2DM group (Table 3) and showed negative correlations with 5mC levels (Table 4), suggesting that sustained inflammation may contribute to global DNA hypomethylation. In line with our results, Cucoreanu et al. [37] reported a link between elevated CRP and reduced global DNA methylation in obese individuals. Taken together, these findings suggest that persistent oxidative stress and inflammation in poorly controlled T2DM may contribute to epigenetic remodeling, and that such molecular changes are reflected in accessible blood-based redox, inflammatory and epigenetic biomarkers.

Emerging evidence suggests that nicotinamide N-methyltransferase (NNMT), an enzyme that catalyzes the methylation of nicotinamide, may play a role in the pathophysiology of T2DM and metabolic syndrome. NNMT activity affects both methyl group availability, by consuming SAM, and NAD^+^ biosynthesis, by depleting nicotinamide and limiting the NAD salvage pathway [38,39]. These metabolic effects have been linked to increased oxidative stress, insulin resistance, and enhanced production of proinflammatory mediators such as TNF-α, IL-6, and CRP [40]. In this context, elevated NNMT expression may represent a shared upstream mechanism contributing to the observed alterations in global DNA methylation, redox imbalance, and chronic inflammation in poorly controlled T2DM. Although NNMT expression was not investigated in our study, its potential relevance to the epigenetic landscape warrants further exploration. Notably, several NNMT inhibitors have already been developed, some of which show promise in restoring metabolic balance and attenuating inflammation and oxidative stress [41,42,43]. The inclusion of NNMT-related pathways in future epigenetic and biomarker studies may offer new mechanistic and therapeutic perspectives.

At present, DNA methylation is considered an integral part of the “metabolic memory” concept, which has been proposed to explain the long-term effects of glycemic control on the risk of diabetes complications. Metabolic memory refers to the phenomenon in which past episodes of hyperglycemia leave lasting effects on cellular function and gene expression, even after glycemic control is improved [44]. Our findings, which demonstrate significantly altered DNA methylation patterns in poorly controlled T2DM patients and their association with redox and inflammatory biomarkers (Table 4), are consistent with the assumption of Dong et al. [44] that the persistence of metabolic memory can lead to sustained oxidative stress and inflammation, further exacerbating epigenetic modifications and promoting the progression of diabetic complications.

Our study has several limitations. First, the relatively small sample size and the limited number of patients within subgroups restrict the strength of conclusions related to specific complications. Second, the cross-sectional design precludes assessment of causality or temporal dynamics, thereby limiting the interpretation of global DNA methylation as a potential predictor of disease progression. Longitudinal studies with repeated measurements are needed to confirm these associations over time. Third, since F-AGEs represent only a fraction of the total AGE pool, sole reliance on F-AGEs may not fully capture the complexity of AGE-related oxidative damage and may partially explain the lack of correlation with DNA methylation levels. Finally, the potential confounding effects of medication use among patients cannot be excluded. To better understand the interplay between global DNA methylation, oxidative stress and inflammation in poorly controlled T2DM, further studies with larger cohorts, longitudinal follow-up and more comprehensive assessment of AGEs and other confounding factors are needed.

## 4. Materials and Methods

### 4.1. Study Participants

This multicentric study involved 107 patients with poorly regulated T2DM and 56 controls. The control subjects were healthy volunteers employed at the Faculty of Pharmacy, University of Belgrade, while the patients were recruited from the Clinical Hospital Center Zvezdara, the Clinical Hospital Center Dr. Dragisa Misovic—Dedinje, in Belgrade, and the General Hospital in Pozarevac. Patients were selected on the basis of glycated hemoglobin (HbA_1c_) values (>7%), indicating poorly controlled diabetes. The collected data encompassed demographic and clinical variables, including age, gender, weight and height, smoking habits, duration of diabetes, hypertension, medications, and presence of complications. The patients’ data were collected from medical records by the attending clinicians. The data of the control subjects were collected through interviews. Hypertension was defined as systolic blood pressure of ≥130 mmHg and/or diastolic blood pressure of ≥80 mmHg, or ongoing antihypertensive treatment. Obesity was defined as a body mass index (BMI) of ≥30 kg/m^2^. BMI was calculated for each patient by dividing weight in kilograms by the square of height in meters.

In the T2DM group, 27 patients (25.2%) received insulin, 37 patients (34.6%) were treated with oral hypoglycemic agents (OHAs), and 31 patients (29.0%) received a combination of insulin and OHAs. The remaining 12 patients (11.2%) reported no prior use of antidiabetic therapy. In particular, of the 68 patients receiving OHAs, 29 patients (42.65%) used metformin, 6 patients (8.82%) used sulfonylureas, 3 patients (4.41%) used SGLT2 inhibitors, one patient (1.47%) used dipeptidyl peptidase-4 (DPP-4) inhibitors, and one patient (1.47%) used thiazolidinediones, while 28 patients (41.2%) used two or more types of OHAs. In addition, 19 patients (17.8%) received statins and 66 patients (61.7%) received antihypertensives.

The study adhered to the ethical principles outlined in the Declaration of Helsinki and received approval from the Ethics Committees of all three hospitals: Clinical Hospital Center Dr. Dragisa Misovic—Dedinje (protocol number: 01-5656/24), Clinical Hospital Center Zvezdara (protocol number: 15/10/2020) and General Hospital in Pozarevac (protocol number: 7190/2020) and University of Belgrade—Faculty of Pharmacy (protocol number: 835/2 and 2566/3). Written informed consents were obtained from all study participants.

### 4.2. Laboratory Analyses

The blood samples were collected after overnight fasting. Serum TC, TG, and HDL-C levels were determined using routine laboratory tests (Beckman Coulter, Brea, CA, USA). The LDL-C levels were then calculated by the Friedewald equation. Serum glucose levels were measured enzymatically (Dimension RxL Max, Siemens, Erlangen, Germany), and HbA_1c_ levels were measured using an immunoturbidimetric method (Innova Star, DiaSys; Dimension EXL, Siemens, Erlangen, Germany). The complete blood count (CBC) analysis was performed using the Sysmex XN-1000 analyzer (Sysmex, Kobe, Japan). Leukocyte-based indices were calculated as follows: NLR by dividing the number of neutrophils by the number of lymphocytes, NWR by dividing the number of neutrophils by the number of leukocytes, MLR by dividing the number of monocytes by the number of lymphocytes, MWR by dividing the number of monocytes by the number of leukocytes, LWR by dividing the number of lymphocytes by the number of leukocytes, and the platelet-to-lymphocyte ratio (PLR) by dividing the number of platelets by the number of lymphocytes.

The assessment of redox status included spectrophotometric measurement of pro-oxidant biomarkers: TOS and PAB; oxidative damage biomarkers: IMA and AOPP; and antioxidant defense biomarkers: TAS, SOD activity, and SH-groups content. These measurements were performed according to previously published methods [45] using the ILAB 300 Plus analyzer (Instrumentation Laboratory, Milan, Italy).

The concentration of fluorescent AGEs (F-AGEs) was measured in serum using a rapid spectrofluorimetric detection method, as previously described [46]. In brief, serum samples were diluted 50-fold with phosphate-buffered saline (PBS) and thoroughly mixed. Then, 300 μL of the diluted serum was transferred into black 96-well plates. Absorbance was measured at excitation and emission wavelengths of 355 nm and 460 nm, respectively, using a Spectrofluorimeter Appliskan (Thermo Fisher Scientific, Vantaa, Finland) with SkanIt Software 2.3 RE for Appliskan. PBS was used as a blank control. Fluorescence intensity was expressed in arbitrary units per milliliter of serum (AU/mL).

### 4.3. Assessment of Global DNA Methylation

Global DNA methylation was assessed in PBMCs. The analysis of global DNA methylation included several consecutive steps: isolation of PBMCs from whole blood, extraction of DNA from PBMCs, hydrolysis of DNA to nucleosides, and quantification of nucleosides using ultra-high-performance liquid chromatography with diode array detection (UHPLC-DAD).

PBMCs were isolated from whole blood using the optimized Ficoll density gradient method. Briefly, whole blood was collected with the K_2_EDTA anticoagulant and centrifuged at 1800 g for 10 min. The white blood cells were then mixed with an equal volume of PBS. After several centrifugation and washing steps with PBS, the PBMCs were stored at −80 °C until use. DNA was extracted from the PBMCs using the FlexiGene DNA Kit (Qiagen, Hilden, Germany), according to the manufacturer’s instructions. DNA quantification was performed using the Spectrostar NANO LVis Plate (BMG Labtech, Ortenberg, Germany). DNA purity and integrity were validated by measuring absorbance at 260 nm and 280 nm, comparing the absorbance ratios at these wavelengths, and by agarose gel electrophoresis. Hydrolysis of 6 µg DNA per sample was carried out using the DNA Degradase PlusTM (Zymo Research, Irvine, CA, USA), according to the manufacturer’s instruction. All hydrolysates were freshly prepared prior to UHPLC analysis.

The reversed-phase chromatography method was adapted from Crescenti et al. [47] for use with small sample volumes and for faster separation. UHLPC was performed on an UltiMate™ 3000 Basic Automated System (Thermo Fisher Scientific, Waltham, MA, USA) with a ZORBAX Eclipse Plus C18 column (Agilent Technologies, Santa Clara, CA, USA) and a ZORBAX Eclipse X DB-C18 HPLC guard column (Agilent Technologies, Santa Clara, CA, USA) using methanol/water as the mobile phase. The gradient chromatography conditions that provided the best separation were as follows: for the first 10 min, 5% methanol was used, followed by a gradual increase to 100% methanol over the next 5 min. From 15 to 25 min, the flow was maintained with 100% methanol. Then, over the next 5 min (up to 30 min), methanol was decreased back to 5%, and this concentration was maintained until the end of the analysis at 60 min. The flow rate was kept constant at 1 mL/min. The column was heated to 30 °C. Calibration curves for 5mC and deoxycytidine (dC) were constructed, based on serial dilutions of commercially available standards. In hydrolyzed DNA, the percentage of global DNA methylation was calculated as the percentage of 5mC in total cytosine (5mC + dC), where concentrations were derived from the area under the curve (AUC) of HPLC peaks, using Chromeleon™ Chromatography Data System software v7.2.4.8179 (Thermo Fisher Scientific, Waltham, MA, USA).

### 4.4. Statistical Analysis

Data were presented as means ± standard deviations or medians with interquartile ranges for continuous variables and as relative or absolute frequencies for categorical variables. Comparisons between groups were performed using Student’s *t*-tests or Mann–Whitney U tests for continuous variables and Chi-square tests for categorical variables. Spearman correlation coefficients were calculated to examine associations between variables. Multiple linear regression analysis was employed to identify independent predictors of global DNA methylation levels. PCA with varimax-normalized rotation was performed to extract factors, which were than included in a binary logistic regression analysis to test for a possible independent association with DNA hypomethylation in T2DM, defined as 5mC levels below the 25th percentile (5mC < 3.40%). In PCA analysis, an eigenvalue >1 was used to extract factors, and variables with factor loadings >0.5 were considered for interpretation. Statistical analysis was performed using PASW^®^ Statistic v.18 (Chicago, IL, USA). A significance level of *p* < 0.05 was considered statistically significant.

## 5. Conclusions

Our research revealed that patients with poorly controlled T2DM have global DNA hypomethylation that worsens with increasing age, longer duration of diabetes, and the presence of microvascular complications, particularly diabetic foot ulcers. However, the relatively small number of participants in subgroups with microvascular complications limits the strength of conclusions that can be drawn regarding specific complications. Additionally, we identified the “pro-oxidant factor” as a significant predictor of DNA hypomethylation in poorly controlled T2DM. Overall, these results highlight the complex interplay between epigenetic modifications, inflammation, and oxidative stress in poorly controlled T2DM. Lower global DNA methylation levels in T2DM patients, particularly in those with longer disease duration and severe complications, point out the potential of global DNA methylation as a biomarker for disease progression. Future longitudinal studies are essential to validate the predictive potential of 5mC for disease progression.

## Figures and Tables

**Figure 1 ijms-26-06716-f001:**
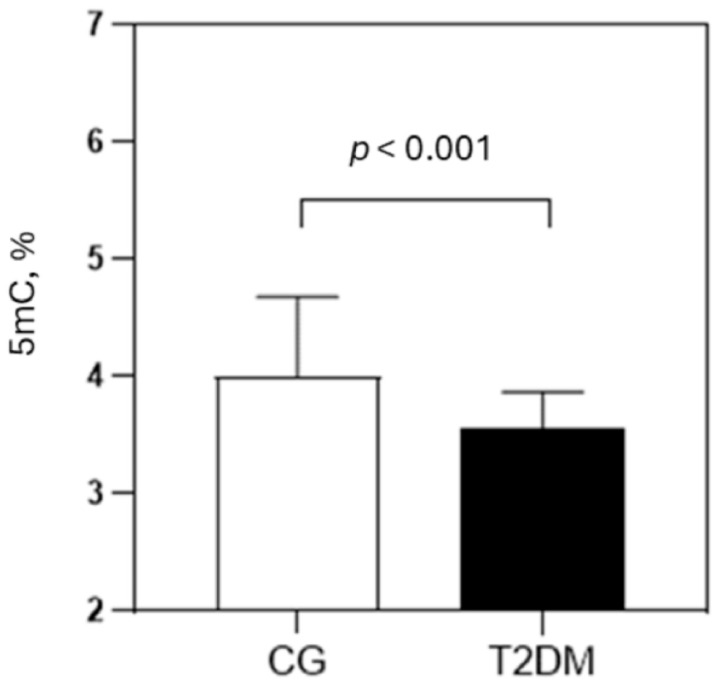
Global DNA methylation in CG and T2DM patients. Abbreviations: CG—control group; T2DM—type 2 diabetes mellitus; 5mC—5-methylcytosine. Data were shown as mean ± standard deviation and compared by Student’s *t*-test.

**Table 1 ijms-26-06716-t001:** Demographic and laboratory data of study participants.

Parameters	CG (N = 56)	T2DM (N = 107)	*p*
Age, years	52.6 ± 9.3 (57)	62.0 ± 11.3 (62.5)	<0.001
Gender, m/f	20/36	60/47	0.021
BMI, kg/m^2^	25.6 ± 3.6 (25.9)	31.5 ± 6.8 (29.6)	<0.001
Obesity, %	7.1	49.5	<0.001
Systolic blood pressure, mm Hg	122.4 ± 20.3 (124)	133.9 ± 20.7 (130)	0.001
Diastolic blood pressure, mm Hg	80.5 ± 10.8 (85)	80.5 ± 11.7 (80)	0.990
Hypertension, %	35.7	88.7	<0.001
Smoking, %	5.4	38.3	<0.001
Glucose, mmol/L	5.23 ± 0.61 (5.20)	10.60 ± 4.08 (10.05)	<0.001
HbA_1c_, %	3.4 ± 1.1 (3.2)	9.9 ± 1.9 (10.0)	<0.001
TC, mmol/L	6.17 ± 1.14 (6.27)	4.61 ± 1.21 (4.70)	<0.001
LDL-C, mmol/L	3.88 ± 1.01 (4.29)	2.64 ± 0.99 (2.70)	<0.001
HDL-C, mmol/L	1.65 ± 0.39 (1.50)	1.06 ± 0.39 (1.00)	<0.001
TG, mmol/L ^#^	1.19 (0.98–1.54)	1.75 (1.10–2.65)	<0.001

Abbreviations: BMI—body mass index; HbA_1c_—glycated hemoglobin; TC—total cholesterol; LDL-C—LDL-cholesterol, HDL-C—HDL-cholesterol, TG—triglycerides. Data were shown as mean ± standard deviation and as medians (in parentheses) and as absolute or relative frequencies. ^#^ Data were shown as median (Q1–Q3).

**Table 2 ijms-26-06716-t002:** Global DNA methylation according to the clinical characteristics of T2DM patients.

Category of T2DM Patients	N	5mC, %	*p*
Gender			
Male	60	3.56 ± 0.27	0.912
Female	47	3.56 ± 0.35	
BMI			
<25 kg/m^2^	54	3.53 ± 0.33	0.246
≥25 kg/m^2^	53	3.60 ± 0.29	
Diabetes duration			
<10 years	69	3.62 ± 0.33	0.027
≥10 years	38	3.48 ± 0.24	
Smoking			
No	66	3.53 ± 0.28	0.090
Yes	41	3.64 ± 0.34	
CVD			
No	59	3.58 ± 0.34	0.676
Yes	48	3.55 ± 0.27	
Microvascular complications			
No	41	3.63 ± 0.36	0.060
Yes	66	3.51 ± 0.26	
Neuropathy			
No	56	3.60 ± 0.36	0.236
Yes	51	3.52 ± 0.24	
Retinopathy			
No	99	3.55 ± 0.32	0.324
Yes	8	3.67 ± 0.16	
Nephropathy			
No	90	3.58 ± 0.31	0.097
Yes	17	3.45 ± 0.30	
Diabetic foot ulcers			
No	82	3.60 ± 0.32	0.037
Yes	25	3.45 ± 0.26	

Data are shown as mean ± standard deviation.

**Table 3 ijms-26-06716-t003:** Inflammatory and redox biomarkers in the studied groups.

Biomarkers	CG (N = 56)	T2DM (N = 107)	*p*
Inflammatory biomarkers			
hsCRP, mg/L	0.30 (0.10–0.85)	5.80 (1.85–13.55)	<0.001
NLR	1.95 (1.35–2.17)	2.19 (1.63–2.93)	0.013
NWR	0.60 (0.53–0.63)	0.60 (0.54–0.67)	0.407
MLR	0.16 (0.13–0.20)	0.24 (0.20–0.31)	<0.001
MWR	0.05 (0.04–0.06)	0.06 (0.05–0.08)	<0.001
LWR	0.31 (0.29–0.38)	0.28 (0.23–0.34)	0.001
PLR	106.90 (89.75–138.80)	111.00 (91.10–150.70)	0.326
Pro-oxidant biomarkers			
TOS, μmol/L	4.5 (4.0–6.0)	6.60 (4.7–8.0)	<0.001
PAB, U/L	74.24 (60.30–84.10)	88.33 (67.60–127.20)	<0.001
Oxidative damage biomarkers			
IMA, AU	0.28 (0.20–0.33)	0.55 (0.47–0.62)	<0.001
AOPP, μmol/L	19.20 (18.25–27.35)	48.90 (36.60–58.90)	<0.001
Antioxidant defense biomarkers			
TAS, μmol/L	1290.0 (1217.0–1384.5)	962.00 (714.0–1120.0)	<0.001
SH-groups, mmol/L	0.35 (0.31–0.39)	0.25 (0.19–0.31)	<0.001
SOD, U/L	133 (124–141)	136 (126–142)	0.162

Abbreviations: hsCRP—high-sensitivity C-reactive protein; NLR—neutrophil-to-lymphocyte ratio; NWR—neutrophil-to-white blood cell ratio; MLR—monocyte-to-kymphocyte ratio; MWR—monocyte-to-white blood cell ratio; LWR—lymphocyte-to-white blood cell ratio; PLR—platelet-to-lymphocyte ratio; TOS—total oxidant status; PAB—pro-oxidant antioxidant balance; IMA—ischemia-modified albumin; AOPP—advanced oxidation protein products; TAS—total antioxidant status; SH-groups—total sulfhydryl groups; SOD—superoxide dismutase. Data are shown as median (Q1–Q3).

**Table 4 ijms-26-06716-t004:** Correlations of 5mC level with inflammatory and redox biomarkers.

	5mC, %	
	r	*p*
Inflammatory biomarkers		
hsCRP, mg/L	−0.388	<0.001
MLR	−0.174	0.035
Redox biomarkers		
PAB, U/L	−0.343	<0.001
IMA, AU	−0.412	<0.001
AOPP, μmol/L	−0.214	0.009
TAS, μmol/L	0.386	<0.001
SH-groups, mmol/L	0.316	<0.001

Abbreviations: hsCRP—high-sensitivity C-reactive protein; MLR—monocyte-to-lymphocyte ratio; PAB—pro-oxidant–antioxidant balance; IMA—ischemia-modified albumin; AOPP—advanced oxidation protein products; TAS—total antioxidant status; SH-groups—total sulfhydryl groups.

**Table 5 ijms-26-06716-t005:** Factors extracted in PCA.

Factors	Variables Included in the Factor	Loadings of the Variables	Factor Variability, % (Total Variance: 58%)
Proinflammatory factor	hsCRP, mg/L SOD, U/L NLR PAB, U/L IMA, AU	0.774 −0.689 0.621 0.607 0.576	18
Pro-oxidant factor	TOS, μmol/L AOPP, μmol/L BMI, kg/m^2^	0.708 0.643 0.599	12.5
Ageing factor	Age, years Diabetes duration, years	0.757 0.723	10.5
Hyperglycemic factor	F-AGEs, U/mL Glucose, mmol/L Gender	0.687 −0.544 0.525	9
Antioxidant factor	SH-groups, mmol/L	0.715	8

Abbreviations: hsCRP—high-sensitivity C-reactive protein; SOD—superoxide dismutase; NLR—neutrophil-to-lymphocyte ratio; PAB—pro-oxidant-antioxidant balance; IMA—ischemia-modified albumin; TOS—total oxidant status; AOPP—advanced oxidation protein products; BMI—body mass index; F-AGEs—fluorescent advanced glycation end-products; SH-groups—total sulfhydryl groups.

**Table 6 ijms-26-06716-t006:** Logistic regression analysis for prediction of DNA hypomethylation by PCA-derived factors.

Predictors	B (SE)	Wald Coefficient	OR (95% CI)	*P*
Proinflammatory factor	−0.777 (0.476)	3.074	0.460 (0.193–1.096)	0.080
Prooxidant factor	0.830 (0.377)	4.863	2.294 (1.097–4.798)	0.027
Ageing factor	−0.120 (0.346)	0.120	0.887 (0.450–1.749)	0.730
Hyperglycemic factor	0.278 (0.410)	0.460	1.321 (0.591–2.949)	0.498
Antioxidant factor	0.173 (0.406)	0.181	1.188 (0.537–2.631)	0.671

Abbreviations: SE—standard error; OR—odds ratio; CI—confidence interval.

## Data Availability

The data presented in this study are available on request from the corresponding author. The data are not publicly available due to privacy and ethics considerations.

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
