# Peer review of "Global DNA Methylation in Poorly Controlled Type 2 Diabetes Mellitus: Association with Redox and Inflammatory Biomarkers"

_ijms, 2025, doi:10.3390/ijms26146716_

Round 1

Reviewer 1 Report

Comments and Suggestions for Authors

Peer Review Report

Global DNA methylation in poorly controlled type 2 diabetes mellitus: association with redox and inflammatory biomarkers

This is a well-designed, detailed, and methodologically sound study that investigates the relationship between global DNA methylation and oxidative stress/inflammation in patients with poorly controlled type 2 diabetes mellitus (T2DM). The study is timely, addresses a clinically and mechanistically relevant question, and is supported by solid statistical analyses and appropriate methodology. The manuscript is clearly written, with logical structure and adequate referencing.

  1. The study addresses a significant gap in understanding the epigenetic regulation of T2DM, particularly focusing on patients with poor glycemic control.
  2. The integration of redox and inflammatory biomarkers with methylation patterns is a notable strength and adds mechanistic insight.
  3. The case-control design is appropriate for the study objectives.
  4. Inclusion of a wide panel of biochemical and hematological markers enhances the depth of analysis.
  5. The use of UHPLC-DAD for global 5mC quantification is a strength due to its precision.
  6. Adequate description of sample collection, processing, and analysis adds to reproducibility.
  7. The use of multivariate regression and PCA to identify independent predictors is commendable.
  8. Adjustments for age and gender, as well as logistic regression based on PCA-derived factors, strengthen the conclusions.

Major Concerns

    • Although well-analyzed, the relatively small number of participants, especially in subgroups (e.g., only 8 with retinopathy), limits the strength of conclusions regarding specific complications. This limitation is acknowledged but deserves stronger emphasis in both the Discussion and Conclusion.
    • The manuscript rightly identifies that fluorescent AGEs represent only a portion of the total AGEs. However, relying solely on F-AGEs as a proxy may be misleading and may partially explain the lack of correlation with DNA methylation. It is advisable to include this limitation more prominently in the
    • The cross-sectional nature limits inference on causality or temporal relationships. While acknowledged, it should be more clearly mentioned that longitudinal studies are essential to validate the predictive potential of 5mC for disease progression.

    • In the results section clarify that logistic regression was used with the PCA-derived factor scores as independent variables to avoid misinterpretation.
    • Include a brief note on the lack of gender-specific analysis in discussion. Given the significant difference in gender distribution, exploring sex-specific effects would strengthen the analysis or at least merit a comment.
    • Language and grammer is generally strong, but a few typographical errors and spacing issues should be corrected.
    • Consider light proofreading for hyphenation artifacts.
    • Ensure that figure legends clearly explain abbreviations and statistical tests used.
    • For tables, consider highlighting statistically significant values for quick visual interpretation.
    • Cite the article Islam S, Moinuddin, Mir AR, Arfat MY, Alam K, Ali A. Studies on glycoxidatively modified human IgG: Implications in immuno-pathology of type 2 diabetes mellitus. Int J Biol Macromol. 2017 Nov;104(Pt A):19-29. doi: 10.1016/j.ijbiomac.2017.05.190. Epub 2017 Jun 3. PMID: 28583871 in the introduction section after the sentence "Additionally, persistent hyperglycemia leads to the formation of advanced glycation end-products (AGEs), which may further increase ROS production through their interaction with receptors for AGEs (RAGE) [9]. Before citing add a sentence “ Glycoxidative modifications of proteins such as IgG have been implicated in the immunopathology of T2DM, supporting the pathological relevance of AGEs in the disease process [Islam et al., 2017]."

Recommendation

Minor Revision

The manuscript is scientifically sound and makes a valuable contribution to the field of diabetes and epigenetics. Minor revisions, particularly to address limitations more thoroughly and enhance clarity, will strengthen the impact and readability of the manuscript.

Reviewer 2 Report

Comments and Suggestions for Authors

The manuscript “Global DNA methylation in poorly controlled type 2 diabetes mellitus: association with redox and inflammatory biomarkers” is a research article

The manuscript could be of interest for the readers, the study design is appropriate and the findings are interesting. However, there are some major concerns that prevent the publication of the manuscript in this form. Authors are asked to address the following concerns:

  1. Some typos are present e.g. line 71 “in-flammation”. Please revise thoroughly the text.
  2. The abstract could be improve better underlying the background and aim of this study.
  3. I strongly suggest a better representation of figure 1. The standard deviation is excessively small compared to the graph bars. A better representation (e.g. with GraphPad) will improve the readability of the manuscript.
  4. Please report the exact P values.
  5. There is a lack of mechanistic insight between global DNA methylation and association with redox and inflammatory biomarkers.
  6. In table 1 is important to show also the median, not only the mean value.
  7. The findings of this manuscript should be associated with the role of Nicotinamide N-methyltransferase (NNMT) in diabetes and metabolic syndrome. By catalyzing the methylation of nicotinamide, NNMT influences critical cellular pathways that govern energy balance, redox state, and insulin sensitivity (PMID: 38919254). Moreover, its high expression can deplete S-adenosylmethionine (a methyl donor) and can reduce NAD levels due to depletion of nicotinamide preventing the NAD salvage pathway (PMID: 39273039). This metabolic disruption not only promotes fat accumulation and systemic insulin resistance but also heightens oxidative stress and alters inflammatory signaling. Elevated NNMT activity has been linked to increased production of reactive oxygen species (ROS) and upregulation of inflammatory biomarkers such as TNF-α, IL-6, and C-reactive protein (PMID: 20110558), all of which contribute to pancreatic β-cell dysfunction and chronic metabolic inflammation. Importantly, several NNMT inhibitors have already been developed and show promise as therapeutic strategies (PMID: 34572571; PMID: 34704059; PMID: 34424711). Authors are encouraged to investigate this aspect or, at least, to include a discussion about this theme in the manuscript.

Round 2

Reviewer 2 Report

Comments and Suggestions for Authors

The manuscript has been improved, all raised concerns have been addressed, thus the manuscript can be accepted for publication.